# Small Particles, Big Effects: The Interplay Between Exosomes and Dendritic Cells in Antitumor Immunity and Immunotherapy

**DOI:** 10.3390/cells8121648

**Published:** 2019-12-16

**Authors:** Bruno Deltreggia Benites, Marisa Claudia Alvarez, Sara Teresinha Olalla Saad

**Affiliations:** Hematology and Transfusion Medicine Center, University of Campinas, Campinas 13083-970, Brazil; marisacalvarez@yahoo.com (M.C.A.); sara@unicamp.br (S.T.O.S.)

**Keywords:** dendritic cells, exosomes, cancer, immunotherapy

## Abstract

Dendritic cells play a fundamental role in the antitumor immunity cycle, and the loss of their antigen-presenting function is a recognized mechanism of tumor evasion. We have recently demonstrated the effect of exosomes extracted from serum of patients with acute myeloid leukemia as important inducers of dendritic cell immunotolerance, and several other works have recently demonstrated the effects of these nanoparticles on immunity to other tumor types as well. The aim of this review was to highlight the recent findings on the effects of tumor exosomes on dendritic cell functions, the mechanisms by which they can lead to tumor evasion, and their manipulation as a possible strategy in cancer treatment.

## 1. Introduction

In recent years, there has been a growing interest in the role of non-neoplastic cells infiltrating the tumor environment, such as mesenchymal cells, endothelial cells, and immune system components. It is becoming increasingly evident that the biology of tumors, including their proliferation, invasion, and spread capabilities, depend largely on the support or repression of these other cell types, especially the cells of the immune system [1].

Tumor cells use various mechanisms to escape the immune system that may directly impact the prognosis of cancer patients and their response to currently available therapies. These actions include features such as attracting immunosuppressive cell populations to infiltrate the tumor microenvironment, as well as modulating normal immune cells to a more permissive and tolerant phenotype polarized for tumor growth and spread [2]. Furthermore, recent studies have shown the possibility of altering tumor growth and preventing its escape mechanisms by blocking or eliminating these dysfunctional immune cells, or by reprogramming their functions to a cytotoxic state [3].

It is currently known that tumor escape mechanisms depend largely on soluble factors acting on intercellular communication, such as the secretion of cytokines and growth factors. In this context, numerous recent studies have demonstrated an important role of extracellular vesicles (mainly exosomes) released in the tumor microenvironment as important modulators of the immune system [4,5]. Therefore, the aim of this review was to depict the effects of exosomes specifically on the performance of dendritic cells (DCs) present in the tumor microenvironment, as well as their role in tumor evasion and treatment response. In addition, we addressed the promising therapeutic approaches involving both the effect of exosomes on the priming of dendritic cells and the paracrine effects of exosomes released by these important antigen-presenting cells themselves.

## 2. Dendritic Cells Functionality in Intratumoral Immune Infiltrate

The discovery of dendritic cells in 1972 was considered a major milestone in understanding the functioning of the immune system and, later, in grasping the immunology of tumors [6]. Their specialized capabilities for antigen capture, processing, and presentation have been progressively described and can be seen in detail in Figure 1. Dendritic cells present a wide tissue distribution, acting as a surveillance system that connects the innate and adaptive immune systems. They are generated through bone marrow precursors and are classified into four general groups: conventional DCs (cDC), plasmocytoid DCs (pDC), monocyte derived DCs, and Langerhans cells. cDCs are further classified according to their tissue location, surface markers, and more recently by the expression of specific transcription factors as well [7].

The elemental function of DCs is to prime and activate naive T cells for an adaptive immune response. In their immature form, they are avidly capable of antigen capture and are characterized by low expression of major histocompatibility complex (MHC) molecules and co-stimulatory molecules (such as CD80 and CD86) [8]. After recognition of molecular patterns associated with pathogens or other antigenic signals (including the presence of tumor cells), DCs undergo a maturation process with increased expression of MHC and co-stimulators on their surface, as well as releasing cytokines, which are essential for T lymphocyte activation [9,10].

In most tumors, the onset of the T cell-mediated cytotoxic immune response also begins with the presentation of disease-related antigens by the dendritic cells to the cytotoxic CD8+ and helper CD4+ T lymphocytes through molecules of the MHC class I and II, respectively. Following stimulation, naïve CD4+ T cells can be polarized into type 1 helper T cells (TH_1_), which in turn support the generation and proliferation of antigen-specific cytotoxic T lymphocytes. These lymphocytes (through their specific T cell receptor) are capable of recognizing tumor cells that exhibit the specific antigens to which they had been stimulated. Then follows the cytotoxic attack, for which these lymphocytes have different possible tools, such as cell death-associated receptor ligands (such as FasL) or cytolytic proteins released from intracytoplasmic granules (such as granzyme B and perforin) [11,12].

However, tumor clones may develop abilities to escape this defense system and proliferate leading to a new manifest disease or relapse. Tumor evasion can occur by several mechanisms, such as decreased immune recognition through the tumor cells’ production of antigens with lower immunogenicity, reduced expression of MHC molecules leading to lower antigen presentation capacity, and failure to activate effector cytotoxic cells. In addition, tumor cells can stimulate the development of an immunosuppressive tumor microenvironment through cytokine release and proliferation of regulatory T lymphocytes (Tregs) [13].

Precisely because of their inherent plasticity, DCs can become prone to the effects of the immunosuppressive tumor microenvironment: tumor cells can suppress the functions of DCs by polarizing them into an immunotolerant phenotype or by recruiting precursors of immunosuppressive DCs. Several mechanisms have been described to explain these changes, including the effect of various cytokines, such as IL-6 and IL-10, growth factors, and metabolic and oxidative changes [14,15]. More recently, the role of tumor-derived microvesicles has also begun to be unraveled in this process.

## 3. Tumor Immune Evasion: Role of Exosomes

### 3.1. Exosomes in Intercellular Communication: Effects on Immune Response

Exosomes are small vesicles measuring approximately 30 to 150 nm in size secreted by different cell types, which have recently received great attention for their possible role as biomarkers in various pathological conditions [16]. Despite having been previously cited as mere cell debris, recent studies have shown an active role of exosomes in intercellular communication by transporting proteins, RNA, and microRNAs that can significantly alter the function of target cells [17,18]. The exosomes are now known to correspond to intraluminal vesicles of endosomal multivesicular bodies (MVBs), formed after endosome invagination and released into the extracellular space by fusion of the MVBs with the plasma membrane. Due to their cellular origin, these particles contain endosomal pathway-specific marker proteins such as tetraspanins (CD63, CD9, and CD81) and heat shock proteins (HSP70) [19].

The exosomes were first visualized in medium collected from reticulocyte cultures [20]; since then, several cell types have been identified as exosome sources, such as hematopoietic cells, epithelial cells, neurons, and adipocytes among others [21]. Initially, exosomes were suggested to play the role of removing molecules unnecessary for cellular metabolism that were only partially degraded by the lysosomal system [22]. However, as investigations progress, their functions appear to be considerably more complex: platelets secrete coagulation-regulating exosomes [23], extracellular vesicles of cardiac progenitors are capable of inhibiting cardiomyocyte apoptosis following myocardial infarction [24], and astrocyte-derived exosomes decrease neuronal damage caused by hypoxia through in vivo autophagy regulation [25].

It is clear from these observations that since exosomes carry particular profiles of proteins, RNA, and microRNAS but recap the internal content of their source cells, these nanoparticles appear to undergo a process of “selective packaging” as a way of refining and enhancing intercellular communication at distant sites and thus regulating important biological functions [26]. Exosomes are internalized by target cells through direct membrane fusion or endocytosis [27] and act mainly by regulating the expression of specific proteins. These effects can be accomplished through the direct transport of mRNAs to be translated or the delivery of microRNAs that lead to transcriptional repression and consequent genetic silencing [28,29].

Importantly, exosomes as molecular messengers have the potential to modulate several pathological scenarios, such as the maintenance of tumor microenvironments. This is accomplished through different biological processes, mainly those involved in immune responses and that include, for instance, signal transduction and antigen presentation [30,31,32]. In fact, there is growing evidence about the modulation of immune cells functions by exosomes, which can be particularly observed in the case of tumor-derived exosomes. Tumor cells can secrete exosomes capable of attenuating the responses of lymphocytes, macrophages, NK cells, and DCs, as well as promoting the expansion of myeloid-derived suppressor cells (MDSCs), a heterogeneous group of immature myeloid cells involved in states of immunosuppression [29,33].

However, some studies, in contrast, have pointed to a possible role of exosomes in antitumor immunovigilance as carriers of tumor antigens to be loaded into dendritic cells. Wolfers et al. showed that exosomes derived from solid tumors such as colon and breast cancer, when delivered to dendritic cells, lead to the activation of T-cell-mediated immune responses culminating in tumor rejection. Interestingly, there was cross-protection among the various tumors evaluated, pointing to the possibility that these exosomes carry common tumor antigens, which would be easily presented by MHC-I molecules in dendritic cells [34]. A similar approach was tested using exosomes derived from heat-stressed carcinoembryonic antigen (CEA) positive tumor cells. These particles induced DC maturation that culminated in cytotoxic lymphocytic responses and reduced tumor burden [35].

In our laboratory, we initially speculated that exosomes purified from serum of patients with hematologic malignancies could also be useful as an antigenic pulse in the development of new forms of immunotherapy. This seemed promising considering previous results with solid cancers and the fact that exosomes are released containing particular contents of proteins, RNAs, and microRNAs that recap the internal content of the maternal cells, and could thus contain specific antigenic material for the priming of dendritic cells. To evaluate this hypothesis, exosomes from serum of patients with acute myeloid leukemia (AML) and myelodysplastic syndromes (MDS) were purified and used as an antigenic source for DCs in co-cultures with lymphocytes and leukemic K562 cells. Surprisingly, our results demonstrated that incubation of DCs with patients’ exosomes decreased the lysis of target cells, probably corresponding to an immune tumor evasion mechanism in vivo [36].

In fact, because they are responsible for T lymphocyte activation, DCs also play an important role in the sensitive balance between immune response and tolerance. Previous studies have shown that mature DCs can limit effector T-cell responses and promote immune tolerance in response to different signaling molecules such as IL-27 and IL-10 [37,38]. In the case of cancer patients, circulating exosomes could possibly be generated in the tumor microenvironment, also containing immunosuppressive molecules, constituting an effective mechanism for the paracrine induction of tolerance and therefore tumor escape. 

Our group also demonstrated that these DCs stimulated with exosomes from AML patients, despite not altering lymphocyte proliferation rates, led to a marked decrease in INF-γ production by these effector cells, and INF-γ levels were inversely related to CD86 expression in DCs. Therefore, we can speculate that in AML, the exosome-induced suppression of cytotoxicity may, at least in part, be the result of dysregulation in co-stimulatory molecules in DCs, such as CD86, leading to decreased activation of lymphocytes with impaired INF-γ production [36].

Interestingly, a study with a murine model of AML had previously demonstrated the effectiveness of using exosomes as a pulse of DCs [39]. One possible explanation for this discrepancy in relation to human patients may be that the tumor in mice was not autologous, but tumor cells had been injected into these animals. This approach may not necessarily mimic the systemic immunosuppressive environment involved in AML in humans, which may explain why promising preclinical results have not been reproduced in human trials. 

Furthermore, our results are in line with the latest studies published on this topic, consolidating the idea that exosomes participate in the induction of immunotolerance in AML. Hong et al. evaluated AML patients treated with NK cell infusion and could observe the effect of patients’ exosomes present in pretreatment samples. These authors noted that there was a decrease in the cytotoxicity of these NK cells when they were incubated along with the serum containing exosomes that was extracted from patients prior to treatment [40]. Moreover, specifically in relation to dendritic cells, a recent study with prostate cancer patients demonstrated that after incubation of DCs with patients’ exosomes, there was also a significant decrease in the release of inflammatory cytokines and less activation of INF-producing CD8+ lymphocytes [41]. Recent studies have also demonstrated the immunosuppressive potential of extracellular vesicles in gliomas [42], and melanoma-derived exosomes have been shown to inhibit the differentiation of monocytic precursors in dendritic cells, leading to increased TGF-β production and suppression of lymphocyte proliferation [43].

### 3.2. Possible Roles Uncovered for Micrornas and Epigenetics?

Epigenetics describes cellular modifications other than DNA sequence variations that can be heritable and modified by environmental stimuli leading to changes in gene expression. These changes arise due to processes such as DNA methylation, histone modifications, chromatin remodeling proteins, miRNAs, and non-coding RNAs, and have the potential to modify the tumor microenvironment which in turns can influence cancer initiation, proliferation, and metastasis [44,45,46].

DNA methylation is one of the best characterized epigenetic modifications. This process is dynamically regulated through the action of DNA methyltransferases (DNMTs) and ten-eleven translocation (TET) enzymes. DNMTs add methyl groups to certain regions of the genome that contain clusters of CpG sequences (CpG islands); most of them are located upstream of promoters, thus resulting in the silencing of certain genes [47]. In a transplant model, Zhu et al. demonstrated that microvesicles derived from K562 leukemia cells contained BCR-ABL1 mRNA, which can be transferred to mononuclear cells from the normal donor, resulting in the expression of a malignant phenotype [48]. Genomic instability was the main mechanism observed in these cells, and this was achieved by upregulation of methyltransferases and global DNA hypermethylation. Furthermore, the treatment of microvesicles with RNase led to a decrease in DNMT3a and DNMT3b, proving that leukemia-derived microvesicles influenced the methylation pattern of target cells via transmission of microvesicular RNA. Thus, the microvesicular cargo derived from neoplastic cells may deliver enzymes involved in methylation and/or demethylation to recipient cells, inducing changes in the expression of tumor-related genes, which in turn leads to accelerated tumor proliferation and metastasis. 

Chromatin structure is also crucial for gene expression regulation and may be modulated by histone modifications [49]. However, the role of extracellular vesicles in posttranslational histones modifications remains controversial. Sharma et al. used bioinformatic analysis and observed an impressive overlap between genes relevant to transgenerational epigenetic inheritance and the cargo of exosomes released by different cell types, including cancer cells. Among such genes were those related to histone acetylation, deacetylation, ubiquitination, and other histone modifications, indicating that exosomal mRNAs and proteins may directly or indirectly participate in the response to the environmental exposure and epigenetic modification [50]. Importantly, we may speculate that these changes in mononuclear cells may have some specific effects on dendritic cells, which should undergo investigation. 

MicroRNAs (miRNAs) are small noncoding RNAs (17–25 nucleotides long) that also control gene expression by promoting degradation or repressing translation of target mRNAs. miRNAS are seen as master regulators, efficiently tuning fundamental cellular processes such as proliferation, apoptosis, and development [51]. The biosynthesis of miRNAs is a multistep process that starts in the nucleus following transcription and continues to the cytoplasm, where the mature miRNA molecule exerts its main functions [52].

Owing to their relatively small size, miRNAS are the most abundant RNAs in exosomes [53], and novel functions for exosomal miRNAs are being revealed in the effectuation of immune responses. For instance, miR-21 and miR-29a can act as ligands to toll-like receptors (TLR) in immune cells, triggering a TLR-mediated prometastatic inflammatory response that ultimately leads to tumor growth and metastasis [54]. Exosomes isolated from peripheral blood of systemic lupus erythematosus (SLE) patients have recently been demonstrated to have the ability to stimulate the secretion of INF-α by plasmocitoid dendritic cells (pDC), and the authors could show that this effect was related to the microRNAs isolated from these exosomes. Interestingly, synthetic microRNAs containing an IFN induction motif could also activate cytokine secretion and induce pDC maturation, revealing the possibility of manipulating exosomal microRNAs as a potential therapeutic target to be explored also in neoplastic diseases [55].

In summary, extracellular vesicles released by tumoral cells into the microenvironment may influence the phenotype of receptor cells, including their epigenetic status, through delivery of mRNAs and miRNAs. Considering the lack of information on how these specific mechanisms may impact dendritic cells and thus favor tumor proliferation, this remains an exciting field for future scientific exploration. 

## 4. Implications for Therapy

### 4.1. Tumor-derived Exosomes (TEX)

As afore-discussed, among the biological functions previously described for exosomes, we can mention their properties as modulators of the immune system. Depending on their internal content, these nanoparticles may potentiate the activation of CD4+ and CD8+ T lymphocytes [56], and macrophage-derived exosomes are capable of transferring antigens to dendritic cells, thereby increasing the response of CD4+ T cells [57].

Despite the quantitative and qualitative defects observed in the DCs of cancer patients, it is possible to generate and differentiate mature dendritic cells from autologous peripheral blood mononuclear leukocytes through specific culture media supplemented with proinflammatory cytokines [58]. The development and enhancement of these cell culture techniques led to increasing cell yields throughout the 1990s, which allowed the exploration of the pulse of DCs with tumor antigens as a new potential anticancer vaccines [59,60]. The most commonly used conventional protocols employ peripheral blood mononuclear leukocytes that are differentiated ex vivo into monocytes and later on into mature dendritic cells [61,62], which are then pulsed with specific tumor antigens in various forms, such as synthetic peptides [63], tumor nucleic acid [64], and apoptotic bodies [65] among others. 

The first study that tested dendritic cell vaccines in cancer patients was published in 1996 and evaluated the use of tumor peptide-pulsed dendritic cells in four patients diagnosed with follicular B-cell lymphoma. All patients developed detectable antitumor cell responses, and two of the patients presented complete regression of the tumor after the vaccination schedule [66]. Since then, results from both in vitro and in vivo experiments have conclusively demonstrated the safety of these vaccines and their effectiveness in developing a specific immune response against various tumor types [67,68], including hematologic malignancies [69,70].

Considering the ease of obtaining microvesicles from patients’ serum and the possibility of their direct cellular transfection in co-cultures independent of artifacts such as viral vectors and electroporation, several studies aimed to evaluate the use of these microvesicles as an antigenic pulse for dendritic cells. Early animal studies showed that vaccination with DCs pulsed with exosomes derived from leukemic cells resulted in longer survival compared with vaccination with DCs pulsed with tumor lysates or the direct infusion of exosomes only [39,71].

However, as described earlier, these results are contradictory, since the stimulation with exosomes led to immune tolerance of DCs in various types of tumors, such as AML and prostate cancer [36,41]. Therefore, another possibility to be considered is the use of exosomes secreted by dendritic cells themselves after being stimulated with tumor antigens.

### 4.2. Exosomes from Dendritic Cells Pulsed with Tumor Antigens (Dendritic Cell Derived Exosomes–DEX)

The potential use of exosomes as autonomous carriers of therapeutic material is evident when considering their cellular biological origin, allowing these nanoparticles to contain a deformable membrane and cytoplasm-like internal content. This guarantees resistance to rupture during trafficking, and the size allows diffusion even through small spaces such as blood capillary fenestrations, hence the recent interest in engineering and manipulating these exosomes to contain antigenic material for immune stimulation.

However, a simpler, more natural approach with less manipulation involves the use of exosomes secreted by DCs themselves after appropriate antigenic pulse. The rationale for this use comes from the observation that, as DCs themselves, DC-derived exosomes (DEX) retain the expression of important molecules involved in immune activation, such as functional MHC-peptide complexes and co-stimulatory molecules. These particles would still have some advantages over the infusion of the DCs themselves, such as their possible greater tissue dispersion and resistance to immunosuppressive agents [72]. Furthermore, the production process of these vaccines would scarcely differ from the conventional production of vaccines using dendritic cells, solely by adding a purification phase of the nanoparticles from the culture medium, but with the advantage that this acellular material can probably be stored for longer periods.

Initial studies in animal models have shown that exosomes purified from a culture medium in which DCs were pulsed with tumor peptides were capable of triggering potent immune responses leading to tumor regression in murine breast cancer [73]. Exosomes derived from DCs expressing alpha-feto-protein also led to significant antigen-specific responses in hepatocellular carcinoma mainly through remodeling of the tumor microenvironment and immune activation of CD8+ lymphocytes [74].

Despite phase-I clinical trials having confirmed the safety of DEX administration and the feasibility of a clinical scale production, the results in terms of clinical efficacy and specific immune response elicitation have been modest. The first two clinical trials used autologous DCs loaded with peptides of melanoma-associated antigen (MAGE) in advanced lung cancer and melanoma. In the case of lung cancer patients, only one of five patients showed increased T-cell activation evidenced by the enzyme-linked immunospot (ELISPOT) assay [75]. In the case of melanoma, the vaccination schedule led to two patients to achieve disease stability, one minor and one partial response, with some of these patients presenting with progressive disease prior to vaccination. However, as in the other study, no evidence of T-cell response was identified in patients’ peripheral blood [76].

Interestingly, despite this low capacity for lymphocyte activation, these DEX demonstrated the expression of killer cell lectin–like receptor subfamily K member 1 ligands (NKG2D-L) and this fact was probably responsible for the stimulation and activation of NK cells through their NKG2D receptor. In fact, there was an increase in NK cells in peripheral blood and their cytotoxicity was considerably enhanced after treatment [77].

A second generation of DEX vaccines was tested by using IFN-γ to mature DCs, which led to increased expression of co-stimulatory molecules [78]. In the case of lung cancer patients, these IFN-γ-stimulated DCs once again failed to reveal evidence of specific T cell immune responses; however, they stimulated NK cell activation, which correlated with longer progression-free survival [79].

Despite these so far poor results, the mentioned studies have demonstrated the effect of dendritic cell-derived exosomes on the lytic functions of NK cells, and the evolution of these preparations toward greater immunogenicity is hoped to lead to more effective and lasting clinical responses. More details of these techniques and clinical trials are reviewed in the paper by Pitt et al. [72].

### 4.3. Engineering More Immunogenic Nanoparticles for Immunotherapy

Another interesting approach to increase the anti-tumor effect of dendritic cells involves the encapsulation of antigenic material within the exosomes. So far, there has been demonstration of effective internalization of siRNA and RNA in nanoparticles through their electroporation, as well as the possibility of using these nanoparticles as drug carriers, as already demonstrated with doxorubicin [80]. This approach is yet to be tested as a possible tool for inducing expression of tumor-specific antigens within the exosomes so that this material could then be transferred to DCs.

To further improve the immunogenicity of exosomes but in the opposite direction, in 2017, Huang et al. manipulated leukemic cell-derived exosomes to silence TGF-1 expression. With this approach, the pulse of DCs with this material induced a more potent anti-tumor immune response than did non-manipulated exosomes, pointing again to the fact that neoplastic cell-derived exosomes may contain molecules with immunosuppressive capacity that impair their use as an antigenic pulse. Thus, current evidence suggests that exosomes should go through editing processes for a proper anti-tumor immune response [81]. In this sense, another option would be to manipulate these nanoparticles to induce specific protein content. For instance, Segura et al. demonstrated that exosomes derived from mature DCs are able to trigger effective T-cell responses mainly through transfer of MHC class II and intercellular adhesion molecule 1 (ICAM-1) [82]. These findings corroborate the idea that changes in exosome protein composition may alter the immune activation capabilities of exosomes; consequently, editing this content may improve their performance.

Another possible approach is the insertion of specific microRNAs into dendritic cells for enhanced immunogenicity. For instance, exogenous miRNA-155 could be successfully inserted into tumor-cell-derived exosomes and led to increased levels of MHCII and co-stimulatory molecules such as CD83 and CD86 on dendritic cells’ surface after exosomal delivery. This also led to enhanced production of INF-γ and IL-10, demonstrating the possible use of exosomal microRNAs as a maturation stimulus for DCs [83].

## 5. Conclusions and Future Perspectives

The discovery of interactions between tumor cells and the immune system through nanoparticle trafficking has revealed an important factor in the establishment of malignant characteristics such as invasion, angiogenesis induction, and metastasis. Similarly, the purification and manipulation of these nanoparticles open the prospect of a new class of therapeutic tools that can be of great value in treating patients with different types of tumors, using the immune system in a more personalized and possibly less toxic manner. This is a new field of study with exciting possibilities; thus, the development of projects to unravel the biology of these interactions and the implementation of clinical trials should be strongly encouraged.

## Figures and Tables

**Figure 1 cells-08-01648-f001:**
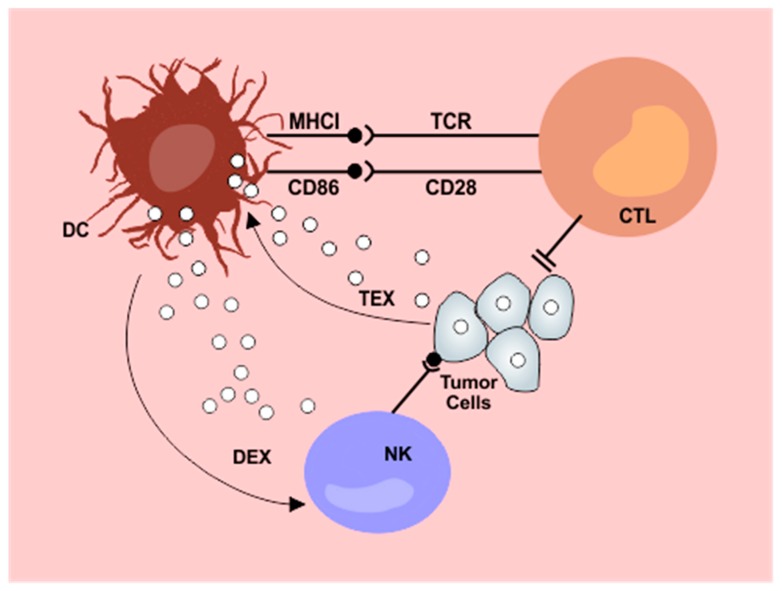
Interplay between dendritic cells and exosomes in the antitumor immunity cycle. Tumor derived exosomes (TEX) are internalized by dendritic cells (DC) resulting in impaired lymphocyte activation. Exosomes released from dendritic cells after contact with tumor antigens (DEX) potentiate NK cytotoxicity. DC: dendritic cells; NK: natural killer cells; CTL: cytotoxic T lymphocytes; DEX: dendritic cell derived exosomes; TEX: tumor derived exosomes; MHCI: major histocompatibility complex I.

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
