# Peer review of "Small Particles, Big Effects: The Interplay Between Exosomes and Dendritic Cells in Antitumor Immunity and Immunotherapy"

_cells, 2019, doi:10.3390/cells8121648_

Round 1

Reviewer 1 Report

The Authors have reviewed the novel and controversial aspect of tumor immunotherapy with the using of in vitro modified DCs. That approach considers the usage of exosomes as tumor antigens mediators for DCs maturation. The presented literature showed minimal impact of such procedure on clinical parameters, however, the discovery of exosomes as new potential option is very exciting and we can assume that this experimental branch will develop rapidly in the nearest future. In my opinion, the engagement of exosome-like vesicles for the transport of therapeutic agents exhibiting systemic toxicity is very interesting and promising.

The manuscript is written in very good English, however, some sentences are too long what make them confusing. Additionally, some sentences require minor grammatical correction, for instance:

Lines 33-34: “What has been made clear is that the mechanisms responsible for tumor escape depend largely on soluble factors acting on intercellular communication, such as the secretion of cytokines and growth factors.”

Lines 134-135: “That the cells of the immune system can be affected by exosomes secreted by other cell types has been made evident. “

Lines 199-200: “Chromatin structure is also crucial for gene expression regulation and histone modifications might influence the global gene expression by modulating chromatin configuration.”

Lines 209-210: “More in depth review of these techniques and clinical trials with DEX can be found in the paper by Pitt et al.”

Lines 313-314: “Another interesting approach to increase the anti-tumor effect of dendritic cells would be the encapsulation within the exosomes of material of interest to be used as antigenic pulse.”

In my opinion, the more accurate phrase is “metastasis” instead of “metastization”.

The Authors should explain the abbreviations only once with their first appearance in text.

I would be appreciated if Authors add a short description of how exosomes act on their target cells to the main text, because this issue isn’t obvious for most of readers.

The best, if Authors could attach a colorful figure illustrating exosome formation, releasing and targeting. Such figure would greatly improve the manuscript value.

Are exosomes described in blood of patients with solid tumors, like colorectal, breast or prostate cancers? Was it evaluated? Are these exosomes more suppressive or stimulatory for DC pulsing?

I have no doubts that the Authors put a lot of work to write their manuscript which, after minor grammatical improvement (and potentially attachment of some new paragraphs and figure), should be accepted for publication by Cells.

Reviewer 2 Report

The review from Benites et al. is informative but might be improved before publication.

Specifically:

The sub-chapter on exosome biogenesis and function seems to be general and maybe out of the scope of the review. It should be either put in the context of DCs or removed.  The authors mentioned microRNAs and mRNAs as exosome cargos. They should also discuss protein cargos, especially those identified in DC exosomes. The description of the effects of TEX on immune response seems to be simplistic and misleading. The authors are encouraged to discuss more deeply their original results in the context of the literature showing the opposite effects. The paragraph should include, at least, few lines about the pro-immune effects of TEX derived from different tumor, such as those in lines 282-287.

Some typos and sentences should be checked. In more detail:

The authors should check for typos in the sub-chapter titles (line 133: they wrote intercelular rather than intercellular and line 312: enginnering instead of engineering). Lines 164-166 are not easily readable. Please rephrase it to help the readership.   The authors should check the style of the reference paragraph. Ref 6 includes some text and is not complete.
